# Development of a screening tool to predict the risk of chronic pain and disability following musculoskeletal trauma: protocol for a prospective observational study in the United Kingdom

Alison B Rushton,[1,2] David W Evans,[1,2] Nicola Middlebrook,[1,2] Nicola R Heneghan,[1] Charlotte Small,[2] Janet Lord,[2] Jaimin M Patel,[2] Deborah Falla[1,2]

[1]Centre of Precision Rehabilitation for SpinalPain (CPR Spine), School of Sport, Exercise and Rehabilitation Sciences, College of Life and Environmental Sciences, University of Birmingham, Birmingham, UK
[2]NIHR Surgical Reconstruction and Microbiology Research Centre, University of Birmingham, Birmingham, UK

**Correspondence to**
Professor Deborah Falla;
d.falla@bham.ac.uk

## ABSTRACT

**Introduction** Pain is an expected and appropriate experience following traumatic musculoskeletal injury. By contrast, chronic pain and disability are unhelpful yet common sequelae of trauma-related injuries. Presently, the mechanisms that underlie the transition from acute to chronic disabling post-traumatic pain are not fully understood. Such knowledge would facilitate the development and implementation of precision rehabilitation approaches that match interventions to projected risk of recovery, with the aim of preventing poor long-term outcomes. The aim of this study is to identify a set of predictive factors to identify patients at risk of developing ongoing post-traumatic pain and disability following acute musculoskeletal trauma. To achieve this, we will use a unique and comprehensive combination of patient-reported outcome measures, psychophysical testing and biomarkers.

**Methods and analysis** A prospective observational study will recruit two temporally staggered cohorts (n=250 each cohort; at least 10 cases per candidate predictor) of consecutive patients with acute musculoskeletal trauma aged ≥16 years, who are emergency admissions into a Major Trauma Centre in the United Kingdom, with an episode inception defined as the traumatic event. The first cohort will identify candidate predictors to develop a screening tool to predict development of chronic and disabling pain, and the second will allow evaluation of the predictive performance of the tool (validation). The outcome being predicted is an individual's absolute risk of poor outcome measured at a 6-month follow-up using the Chronic Pain Grade Scale (poor outcome ≥grade II). Candidate predictors encompass the four primary mechanisms of pain: *nociceptive* (eg, injury location), *neuropathic* (eg, painDETECT), *inflammatory* (biomarkers) and *nociplastic* (eg, quantitative sensory testing). Concurrently, patient-reported outcome measures will assess general health and psychosocial factors (eg, pain self-efficacy). Risk of poor outcome will be calculated using multiple variable regression analysis.

### Strengths and limitations of this study

► A comprehensive array of candidate predictive factors will allow for the prediction of chronic and disabling pain following trauma.
► These predictive factors will enable the development and validation of a predictive tool to predict good and poor outcome at 6 months postinjury.
► The prospective design of the study enables control of unwarranted influences and enables a stronger case for inferring causal relationships.
► Identifying predictive factors related to poor outcome of pain and disability outcome will facilitate targeting of effective interventions.
► Other candidate predictors may have been useful to include (eg, vibration thresholds), but consideration of burden to participants of testing and sample size considerations necessitated prioritisation of candidate predictive factors.

**Ethics and dissemination** Approved by the NHS Research Ethics Committee (17/WA/0421).

## INTRODUCTION

Pain is an expected and appropriate experience that usually follows traumatic injury.[1] By contrast, chronic pain and disability are unhelpful but common sequelae of trauma-related injuries.[2] Gaining an understanding of why some people develop chronic and disabling post-traumatic pain is therefore a priority for individual patients, the military and society at large. Notwithstanding, the mechanisms that underlie the transition from acute to chronic disabling post-traumatic pain are not fully understood. Such knowledge would facilitate the development and implementation of a clinical

pathway of care that matches interventions to projected risk of poor recovery, with the aim of preventing poor long-term outcomes. This project stems from advances in knowledge relating to the assessment and management of pain[3] and the quantification of potential predictive factors to inform personalised rehabilitation; identifying which patients to target with rehabilitation and when and how to target them.

Few studies have specifically explored predictive factors for recovery, whether poor or good, following physical trauma. Of those that have psychosocial variables, such as anxiety, depression and post-traumatic stress, have so far been identified as the strongest predictors of outcome.[4–7] However, only a limited number of variables have hitherto been evaluated as potential predictive factors. Indeed, current opinion regarding pain mechanisms[8] suggests that the development of chronic pain and disability cannot be entirely attributable to psychosocial factors. This is consistent with research in primary care that has identified predictive factors for poor outcome across a range of musculoskeletal pain conditions,[9] which include: widespread pain, high functional disability, high pain intensity, long pain duration, high depression/anxiety, presence of previous pain episodes, movement restriction and poor coping strategies. Moreover, developments in the mechanistic understanding of pain[10–12] suggest that other measures (eg, indicators of central sensitisation and inflammatory activity) may have potential predictive utility, especially in an acute trauma population.

## Aims of study

1. Using a unique combination of (1) general patient characteristics including premorbid neuropsychological status, (2) quality of life and physical functioning, (3) psychosocial features, (4) injury characteristics, (5) pain characteristics, (6) quantitative sensory testing and (7) biomarkers, we aim to find a set of predictive factors to identify patients at risk of developing ongoing post-traumatic pain and disability following acute musculoskeletal trauma. This will subsequently inform the feasibility of developing and evaluating a new clinical care pathway of precision rehabilitation that matches interventions to the predicted risk of poor recovery.

## Objectives

(1) Identify predictive factors for poor outcome (chronic pain and disability at 6 months postinjury) following acute musculoskeletal trauma. (2) Develop a predictive model to inform a screening tool to identify the predicted risk of poor recovery (transition from acute post-traumatic pain to chronic pain and disability). (3) Estimate the predictive performance of the screening tool through validation of the model in a separate dataset. (4) Document the clinical course of symptoms at 3 and 12 months following acute musculoskeletal trauma.

## METHODS AND ANALYSIS
### Source of data

The study will be a prospective, observational study of two temporally staggered cohorts of patients with trauma, who are emergency department admissions into a Major Trauma Centre in the United Kingdom, with an episode inception defined as the traumatic event (figure 1). The first cohort will facilitate development of the prediction model to inform the screening tool, and the second will enable validation of the prediction model through evaluation of the predictive performance of the model and tool.[13 14] There will be an interval of at least 6 months between recruitment into the two respective cohorts. The prospective design enables control of unwarranted influences and enables a stronger case for inferring causal relationships. The nature of the study necessitates predictive statistical modelling.[15] This protocol is written in line with the TRIPOD (transparent reporting of a multivariable prediction model for individual prognosis or diagnosis) statement,[16] in which recommendations are given for the reporting of prediction model development and validation.

Self-reported and physical assessment predictive data will be collected at baseline over a period of up to 14 days (or duration of inpatient stay if shorter), which will commence immediately following recruitment. Biomarker data collection will occur throughout the same baseline period, but can commence prior to recruitment providing assent is gained from a legal consultee. The outcome data will be collected at 6 months postinjury (working definition of chronic pain)[17], the point of evaluation of an individual's absolute risk of poor outcome (objectives 1, 2 and 3). In addition, selected data will be measured at 3 and 12 months postinjury to explore the clinical course of recovery following injury in the shorter and longer terms (objective 4).

### Participants

Participants will be recruited from the register of a Major Trauma Centre in the United Kingdom for a period of up to 24 months (planned start date January 2018). All consecutive eligible patients will be approached for recruitment until the sample size is achieved.

### Eligibility criteria

Inclusion criteria: adult patients aged ≥16 years who are admitted to emergency department of the Major Trauma Centre, with their main criteria for admission being acute musculoskeletal trauma within the preceding 14 days, and a capacity to use and understand written and verbal English language and a mental capacity to provide informed consent (eg, no confusion, delirium, severe cognitive impairment or severe mental illness, defined by a score of ≤6 on the Abbreviated Mental Test).[18] The primary reason for including patients with trauma occurring up to 14 days, is to be inclusive of patients who are critically ill and/or with diminished mental capacity initially following their trauma, patients requiring surgery

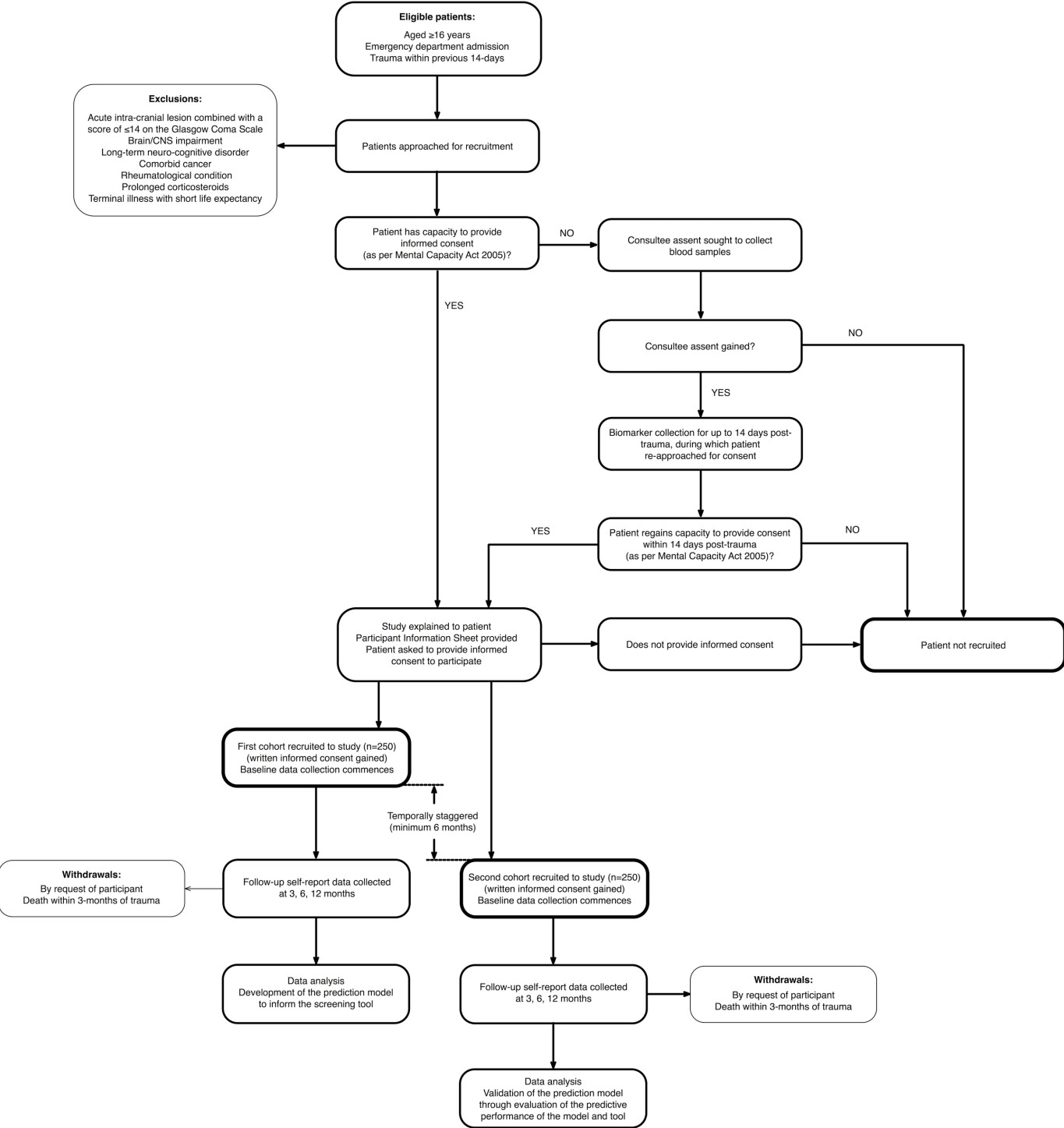

**Figure 1** Study design. CNS, central nervous system.

as a result of the trauma and representative of the broad trauma population.

Exclusion criteria: exclusions will be made where the patient has an acute intracranial lesion (eg, bleed) combined with a score of ≤14 on the Glasgow Coma Scale[19] (a 15-item measure of consciousness impairment with adequate reliability[20] that is routinely taken in patients with trauma at hospital admission), where there is evident brain or central nervous system injury or impairment, long-term neurocognitive disorders (such as

brain tumour, multiple sclerosis, Alzheimer's and Parkinson's diseases and so on), comorbid cancer, the presence of an ongoing rheumatological condition, prolonged use of corticosteroids or terminal illness with short life expectancy.

Withdrawals: participants will be informed that they are free to withdraw from the study at any time, without needing to provide reason. In the event of death within 3 months of being recruited, participants will be automatically withdrawn from the study and the primary predictive

analysis. Baseline data of all withdrawn participants will be kept and compared with those of retained participants to assess for any differences.

### Recruitment

Based on feasibility data (site data from the Trauma Audit and Research Network), it is estimated that at least 1000 eligible patients with trauma will be approachable for recruitment over a 24-month period, and that 50% would be expected to consent to participation. A dedicated team of research nurses will be available to recruit patients 7 days per week (from 0700 to 1930).

Because of impairments resulting from their injuries, some otherwise eligible patients will lack the mental capacity to provide informed consent when first approached to enrol in the study. Recruitment into the study will therefore be undertaken under the guidance and provision of the (UK) Mental Capacity Act 2005 for research in emergency situations. If the patient lacks sufficient capacity to consent, written assent for study participation will be sought from a legal consultee to begin biomarker data collection (blood samples), with informed consent for full recruitment (and subsequent data collection) being sought from the patient only if, and when, they regain sufficient capacity to provide this. If the patient does not regain capacity to provide consent within 14 days of their trauma, they will not be recruited into the study, biomarker data collection will cease and any blood samples already collected will be destroyed before analysis.

Once informed consent is gained and the participant recruited, following a minimum 1 hour lead time after the informed consent process (to reduce patient burden), members of the research team will visit the patient at their bedside to collect baseline self-reported data via questionnaires (table 1). On the next available working day following completion of the questionnaires (again, to reduce patient burden), members of the study team will return to the patient to conduct the first physical (quantitative sensory testing) assessment. At each visit, if deemed necessary, the capacity of the participant will be checked using an Abbreviated Mental Test[18] (a score of ≤6 is indicative of reduced capacity), and asked if they are happy to proceed with data collection.

### Outcome

The outcome for the prediction model is an individual's absolute risk of poor outcome (chronic pain and disability) at 6 months postinjury. Outcome will be measured using the Chronic Pain Grade Scale (CPGS),[21] which combines pain intensity and pain-related disability over the preceding 6 months into a single measure of pain *severity*. The CPGS has previously been used to assess the severity of body-wide chronic pain in general population,[22] primary care[23] and post-trauma[24] populations. Each item of the CPGS relates to at least one of the three categories of the International Classification of Functioning, Disability and Health (ICF)[25]: impairment,

activity limitations and restricted participation. Furthermore, all ICF categories are encompassed by the CPGS.[26] The CPGS is a unidimensional scale, with good internal consistency across different pain populations; Cronbach's alpha of 0.84 to 0.91 in back pain, 0.79 for headache and 0.84 for temporomandibular pain.[21 27] With regards to construct validity, cross-sectional and longitudinal studies of general practice patients have shown high scores on the CPGS, indicating greater chronic pain, to be associated with higher rates of unemployment, greater pain impact scale scores, greater use of opioid analgesics and physician visits, depressed mood and lower self-rated health status.[21 27 28] For convergent validity, the CPGS has been found to have good correlation with equivalent dimensions of the SF-36.[27 28] In terms of responsiveness, changes in score over time in patients with chronic musculoskeletal pain correlated significantly with changes in SF-36 scores.[29] The CPGS has also been shown to have good test–retest reliability in primary care patients with back pain (weighted kappa 0.81, 95% CI 0.65 to 0.98).[27]

Although pain persistence is not used in assigning pain grade, a measure of pain days in the prior 6 months is included in the CPGS.[30] The responses on the remaining seven items are used for computing scores for the three subscales of the CPGS[21]: characteristic pain intensity, disability score and disability points. The characteristic pain intensity score (range: 0–100) is obtained by calculating the mean of three pain intensity measurements: 'at the present time', the 'worst pain' in the preceding 6 months and the 'average' pain over the preceding 6 months. The disability score (range: 0–100) is obtained through the mean ratings of how much the pain has interfered in performing activities of daily living, recreational, social and family activities, and work (including housework) activities in the last 6 months. The disability points are scored 0–3 and are derived from a combination of ranked categories of the number of disability days (the number of days that the respondent was away from usual activities in the preceding 6 months due to pain) and disability score. Based on these scores, the participant's chronic pain and disability status can then be classified into one of the five ordinal categories of chronic pain severity[21]: no pain (Grade 0), low disability and low intensity pain (Grade I), low disability and high intensity pain (Grade II), high disability and moderately limiting intensity pain (Grade III), and high disability and severely limiting intensity pain (Grade IV). As in previous studies, poor outcome will be defined as grade ≥II.[23 31–34]

### Candidate predictors

Candidate predictors have been selected that are: (1) reliable and valid measures of their domain, and (2) have a theoretical association with the development of chronic pain. Both modifiable and non-modifiable candidate predictors are included. Candidate predictors are summarised in table 1, with further detail in the online supplementary file S1. Table 1 details important data that will be measured at 3, 6 and 12 months postinjury to

**Table 1** Summary of data collection at different assessment points

| Domain/Candidate predictor | Measure/data item | Baseline commencing ≤14 days post-trauma | 3-month clinical course | 6-month outcome assessment point/clinical course | 12-month clinical course |
|---|---|---|---|---|---|
| General patient characteristics including premorbid neuropsychological status | | | | | |
| Age | In years | ✓ | | | |
| Gender | Female/male/other | ✓ | | | |
| BMI | Calculated from height and weight measurements | ✓ | | | |
| Education | Highest educational level attained | ✓ | | | |
| Employment status | Full time/part time/not working/retired/student Employed/self-employed | ✓ | ✓ | ✓ | ✓ |
| Circumstance of trauma | Military/civilian | ✓ | | | |
| Previous history of musculoskeletal pain and injury | Patient history data from patient recollection and medical records | ✓ | | | |
| Comorbidity of other health problems | Patient history data from patient recollection and medical records | ✓ | | | |
| Premorbid physical health | Patient history data from patient recollection and medical records | ✓ | | | |
| Premorbid psychological health | Patient history data from medical records and patient recollection (including non-somatic items from the Subjective Health Complaints Inventory)[44] | ✓ | | | |
| Number of days in hospital | Intensive care/ward/total | ✓ | | | |
| Quality of life and physical functioning | | | | | |
| General health | SF-36v2[45] | ✓ | ✓ | ✓ | ✓ |
| Health-related quality of life | EuroQol EQ-5D-5L[46] | ✓ | ✓ | ✓ | ✓ |
| Self-care and mobility during activities of daily living | Barthel Index of Activities of Daily Living,[47] collected from hospital data | ✓ | | | |
| Sleep quality | 11-point (0–10) Numerical Rating Scale, relating to current pain, from 'best possible sleep' to 'worst possible sleep'[48] | ✓ | ✓ | ✓ | ✓ |
| Brain/CNS impairment | Glasgow Coma Scale[19] | ✓ | | | |
| Psychosocial features | | | | | |
| Anxiety and depression | HADS[49] | ✓ | ✓ | ✓ | ✓ |
| Coping strategies applied during a painful experience | CSQ-24[50] | ✓ | ✓ | ✓ | ✓ |
| Fear of movement or fear of injury or re-injury during movement | TSK-11[51] | ✓ | ✓ | ✓ | ✓ |
| Confidence in ability to perform activities despite pain | Pain Self-Efficacy Questionnaire[52] | ✓ | ✓ | ✓ | ✓ |
| Subjective post-traumatic distress | IES-R[53] | ✓ | ✓ | ✓ | ✓ |
| Injury characteristics | | | | | |
| Time of injury/incident | Hospital record data | ✓ | | | |
| Injury location | Adapted pain drawings, based on hospital record data | ✓ | | | |

Continued

**Table 1** Continued

| Domain/Candidate predictor | Measure/data item | Baseline commencing ≤14 days post-trauma | 3-month clinical course | 6-month outcome assessment point/clinical course | 12-month clinical course |
|---|---|---|---|---|---|
| Tissues damaged | Based on imaging data and hospital records / Fractures / Penetrating/non-penetrating injury/both | √ | | | |
| Surgery required | Location and postinjury timing of surgery, based on hospital record data | √ | | | |
| Assisted mechanical ventilation required | Yes/no/duration | √ | | | |
| Severity of injury | Injury Severity Scale[54] | √ | | | |
| Pain characteristics | | | | | |
| Chronic pain severity | Chronic Pain Grade Scale[21] | | | √ | √ |
| Pain intensity | 11-point (0–10) Numerical Rating Scale, relating to current pain, from 'no pain' to 'pain as bad as could be' (collected as part of the Chronic Pain Grade Scale) | √* | √ | √ | √ |
| Pain medication intake (type, dosage and timing) | Medication Quantification Scale,[55–57] based on hospital record data | √* | | | |
| Pain location | Pain drawing | √* | √ | √ | √ |
| Pain extent | Electronic pain drawing[58] | √* | √ | √ | √ |
| Self-reported features of neuropathic pain | painDETECT questionnaire[59] | √* | √ | √ | √ |
| Quantitative sensory testing | | | | | |
| Heat pain threshold | Evaluation of pain threshold using a heat stimulus | √* | | | |
| Cold pain threshold | Evaluation of pain threshold using a cold stimulus | √* | | | |
| Pressure pain threshold | Evaluation of pain threshold using a pressure stimulus | √* | | | |
| Temporal summation | Evaluation of pain intensity in response to repetitive pressure stimuli | √* | | | |
| Biomarkers | | | | | |
| CRP | Serum levels of CRP, a broad indicator of inflammation (via blood analysis) | √† | | | |
| cfDNA | Plasma levels of cfDNA (nuclear and mitochondrial), indicators of tissue damage (via blood analysis) | √† | | | |

*Measurements to be taken repeatedly throughout the baseline period, which will commence immediately following recruitment via informed consent (up to 14 days post-trauma) for a period of up to 14 days (i.e. until a maximum of 28 days post-trauma), while the participant is in hospital.

†Measurements to be taken repeatedly throughout the baseline period, but may be commenced prior to this, subject to assent from a legal consultee.

BMI, body mass index; cfDNA, cell-free DNA; CRP, C reactive protein; CSQ-24, Coping Strategies Questionnaire-2; HADS, Hospital Anxiety and Depression Scale; IES-R, Impact of Event Scale revise; SF-36v2, 36-item Short Form Health Survey, version 2; TSK-11, Tampa Scale of Kinesiophobia, 11-item.

explore the clinical course of recovery following injury in the shorter and longer terms. All data collection will be standardised through protocols and clinical report forms.

## Data handling

Blood samples will be collected through the clinical and research nurse teams, while the participant is in the hospital, and either analysed immediately (C-reactive protein) or securely stored for subsequent analysis (cell-free DNA). Baseline self-reported questionnaires, pain and injury drawings, and physical assessments will be collected by one of three trained assessors from the study team. Inter-rater reliability studies (across two assessors) will first be conducted in both healthy and trauma populations to inform definitive testing protocols. The order of physical assessment data collection will be randomly assigned (using computerised randomisation software) according to the modality of testing and by site, to prevent order effects. Follow-up self-reported questionnaires will be posted to participants at their home addresses; with up to two postal reminders and a telephone call for non-response. All questionnaires will be formatted so that data can be scanned or entered directly into an electronic database. Following data entry, data will be checked by a second researcher for completeness and accuracy. In addition, regular audits of data collection and storage will be performed by an independent study management committee. Participant identifiable information will be securely stored within the hospital, in line with current United Kingdom data protection legislation, and only accessible by the site Principal Investigator and Research Nurse team who will not be involved in data analysis. All outcome measure data will be securely transferred within an anonymised database file to physically secure servers at the University of Birmingham, and stored for a period of 10 years in line with Research Governance procedures. Participants will receive usual care and interventions received will be recorded for descriptive analysis. Anonymised data will be analysed using IBM SPSS Statistics.

## Sample size

In predictive modelling, a larger sample size enables lower bias and variance, and permits the prospective prediction of new observations.[15] The number of predictors will be reduced using exploratory factor analysis. This process will ensure that the sample size provides at least 10 cases per candidate predictor, to adequately power the final regression analysis.[35 36] Data will be collected for an estimated 300 participants per cohort (n=600 total) to allow for withdrawals (primarily expected deaths within the first 3 months) and losses to follow-up, so that final data are available for 250 participants per cohort (n=500 total).

## Statistical analysis methods and management of missing data

For each cohort, potentially eligible patients, numbers examined for eligibility, confirmed eligible, recruited into the study, completing follow-up and analysed will be reported in a flow diagram. Reasons for non-participation,

exclusion, drop-outs and withdrawal (eg, death) will be reported at each stage. Participant characteristics will be descriptively presented. For each variable of interest, the number of participants with missing data will be reported.

For the first cohort to develop the predictive model, an initial exploratory data analysis stage will summarise the data.[15] Correlations between candidate predictive factors will be calculated at baseline. Outcome (CPGS) scores will be dichotomised into good and poor categories as described previously. Data reduction will use exploratory factor analysis to assess factor loading of candidate predictors (summary scores) on poor outcome at 6 months. This will enable the number of candidate predictors entered into the final model to be reduced to 25, which can be supported by the cohort sample of 250. This process reduces the risk of over-fitting the model and the risk of selecting the wrong variables due to correlation between predictor variables (multicollinearity).[37]

Statistical modelling for prediction has been planned *a priori*. To explore the influence of each predictive factor on poor outcome at 6 months, a logistic multivariable regression model will be fitted to the dichotomised outcome scores to calculate low and high risk of poor outcome. Odds ratios for each candidate predictive factor will be reported, adjusted for other factors and account for clustering (eg, level of injury severity). If necessary, multiple imputation[38] will be used to deal with missing outcome data. The characteristics of those patients with and without 6-month data will also be compared, to inform whether patients with no 6-month data were missing at random. Reduced multivariable analyses will be considered if necessary (eg, removing one of two candidate predictive factors that are highly correlated at baseline), to examine the robustness of the conclusions.

## Risk groups and development of the predictive screening tool

The predictive model will be used to develop a risk stratification tool to inform an individual's absolute risk of poor outcome. The stratification tool will inform clinical decision-making for precision rehabilitation. Items will be selected for the model if they are statistically significantly (P<0.05) associated with poor outcome in the logistic regression analysis, and those deemed clinically important to retain using expert opinion (regardless of statistical significance, study steering group) to improve face validity for clinicians and avoid overfitting of the model.[37] The regression model with included predictive factors will be fitted to the data from the first of the two cohorts to obtain a final set of parameter estimates (i.e. alpha and beta terms), which will be used to form the model. An important requirement of the stratification tool is that it should be sufficiently brief to facilitate use in clinical practice. Thus, we will look to simplify the model where possible to facilitate its use, but without important reduction in its predictive ability in terms of calibration and discrimination. For example, if multi-item questionnaire scores are included in the model, then we will

evaluate whether just one of the questionnaire items is sufficient. Ideally, this process will result in a full and simplified model.

### Development versus validation

For validation of the model, data from the second of the two cohorts will be compared with that of the first to enable analysis of the distribution of important variables, inclusive of demographic, predictor and outcome variables. The predictive performance of the screening tool (discrimination, calibration and goodness of fit) will be assessed using data from the second cohort. Data in both cohorts will be consistent in terms of setting, eligibility criteria, outcome and predictors.

## DISCUSSION

There is an urgent need for a robust predictive study to predict the transition from acute to chronic pain in a musculoskeletal trauma population. Using such a comprehensive array of outcome measures will allow the development and validation of a predictive tool to predict development of chronic and disabling pain, and begin the process of identifying appropriate and precision interventions.

The candidate predictors used in this study have been chosen to be as comprehensive as possible, based on current knowledge of pain science. Other candidate predictors were considered (eg, microRNA biomarkers), but their mechanistic functions and temporal progression are not yet sufficiently clear to justify the expense of their inclusion. The combination of patient-reported outcome measures, psychophysical testing and biomarkers that are included are designed to act as surrogates for the four primary mechanisms of pain[8 39 40]: *nociceptive* (injury location, severity and characteristics), *neuropathic* (painDETECT tool and pain extent, *inflammatory* (biomarkers) and *nociplastic* (quantitative sensory testing, painDETECT and pain location and extent). In addition, other patient-reported outcome measures (eg, pain intensity, post-traumatic stress, anxiety and depression, coping and pain self-efficacy) are included as the domains that they measure have been shown to influence prognosis for long-term outcomes in populations with pain in a range of locations.[9 23 24]

Rehabilitation is widely regarded as an important component of post-trauma healthcare[41]; however, the current position of equipoise means that precision rehabilitation has not yet been identified. Understanding mechanisms that underlie the transition from acute to chronic pain is essential to moving beyond this position. Identifying predictive factors related to poor outcome of pain and disability outcome will facilitate targeting of effective interventions. This will inform rehabilitation decision-making and facilitate improvements in clinical and cost-effectiveness of rehabilitation interventions.

Limited research has identified criteria for quality in a predictive model, but authors have identified potential quality issues to ensure methodological rigour.[42] These issues are summarised in table 2 and incorporated into the study design to ensure low risk of bias in development and validation of the predictive model.

### Patient burden and potential distress

The primary ethical concern is to limit distress on participants. As such, to reduce the patient burden when collecting baseline data, the self-reported questionnaires will be administered by members of the study team shortly following obtaining fully informed consent, and physical assessment outcomes will be measured at least 24 hours later. Patients will be asked for consent to not only provide new data for the study, but also for the study team to access information that will have been routinely collected by the hospital staff since the time of admission (eg, nature and circumstances of injury, medical history, medication details and blood test results). This will be fully explained to patients and explicitly detailed in the participant information sheet.

### Mental capacity

Because of the nature of their injuries, the patient's mental capacity will be assessed on admission into hospital and thereafter by clinical staff and/or research nurses. The mental capacity of eligible patients at the time of being approached for recruitment will therefore fall into one of two groups: either they possess or are lacking mental capacity (in accordance with the Mental Capacity Act 2005) to provide informed consent to voluntarily participate in the study.

For patients possessing mental capacity to provide consent, a research nurse or member of the research team will ask if they are interested in participating in the study. If they are interested, a copy of the participant information sheet will be provided (and if necessary read to them) to give them an outline of the study. Following an opportunity to seek additional information and ask questions, the patient will be asked if they wish to provide their written informed consent to participate in the study, at which point a consent form will need to be signed.

On admission to the hospital, an otherwise eligible patient may lack the mental capacity to decide whether to provide consent to participate in a research study (eg, due to the severity of their injuries, because they are arriving intubated and ventilated, or as a side-effect of medication for their injuries). They may or may not regain this capacity during their stay in the hospital. Due to our wish to begin measuring biomarkers as early as possible following the onset of trauma, for some otherwise eligible patients it would be necessary to take blood samples before the patient has regained the capacity to provide informed consent. Using the convention of previous studies in trauma populations,[43] recruitment into the study will be undertaken under the provision and guidance of the Mental Capacity Act 2005 for research in emergency situations, and in accordance with the Declaration of Helsinki. As such, if a patient does not possess this capacity when first approached for recruitment, the research team will request a mandate to collect

**Table 2** Methodological decisions to improve study quality

| Criteria[42] | Methodological decisions to improve quality |
|---|---|
| **Study design** | |
| Inception cohort | ► Clear description of population<br>► Clear description of the participants at baseline |
| Source population | ► Clear description of population<br>► Clear description of sampling frame and recruitment (method and timing) |
| Inclusion and exclusion criteria | ► Clarity of eligibility criteria |
| Prospective design | ► Clarity of study design |
| **Study attrition** | |
| Number of drop-outs | ► Adequate participation rate<br>► Clear description of attempts to collect information on participants who dropped out<br>► Reporting numbers and reasons for loss to follow-up |
| Information provided on method of management of missing data | ► Appropriate methods of imputation of missing data |
| **Predictive factors** | |
| All predictive factors described used to develop the model | ► Clear definition of predictive factors<br>► An adequate proportion of participants has complete data for the predictive factor |
| Standardised or valid measurements | ► The measurement of the predictive factor is reliable and valid<br>► The measurement of the predictive factor is the same for all participants |
| Linearity assumption studied | ► Linearity of data will be reported |
| No dichotomisation of predictive variables | ► Continuous variables will be reported where possible |
| Data presentation all predictive factors | ► Complete data will be presented |
| **Outcome measures** | |
| Description of outcome measures | ► The outcome is clearly defined |
| Standardised or valid measurements | ► The measurement of the outcome is reliable and valid<br>► The measurement of the outcome is the same for all participants |
| Data presentation of most important outcome measures | ► Complete data will be presented |
| **Analysis** | |
| Presentation of univariate crude estimates | ► An appropriate strategy for model building is described<br>► An adequate statistical model described |
| Sufficient numbers of subjects per variable | ► Adequate data will be presented |
| Selection method of variables explained | ► Sufficient data will be presented to enable assessment of the adequacy of the analytic strategy<br>► All results will be reported |
| Presentation of multivariate estimates | ► An appropriate strategy for model building is described<br>► An adequate statistical model described |
| **Clinical performance/validity** | |
| Clinical performance | ► Clinical performance of the model will be reported |
| Internal validation | ► Internal validation will be reported |
| External validation | Not a focus of this study |

blood samples from a legal consultee. This mandate can continue until the patient gains sufficient capacity to make an informed decision as to whether they wish to provide consent or not. We will use this mandate up to 14 days from the date of the trauma. If the patient does not regain capacity within 14 days following the trauma, or if informed consent is not provided by the patient once capacity to do so is regained, any samples collected will be destroyed before any non-clinical biomarker analysis (i.e. cell-free DNA) is performed. Furthermore, only once informed consent has been gained from the patient would the research team proceed to collect any

self-reported questionnaire or physical assessment data. The legal consultee can either be a 'personal consultee', for example, family member, or a 'nominated consultee', for example, intensive care consultant. Once a consultee (personal or nominated) has been identified, they will be provided with the participant information sheet, to inform them about the study. The consultee will be asked if they feel participating in the study would be something to which the patient would agree or object to. If, in their opinion, the patient would agree to participating in the study, the consultee will be asked to sign a declaration form, and the research team can begin the schedule of blood sample collections. If, at any time prior to the patient regaining capacity, the consultee decides to withdraw assent, then no further samples will be collected until the patient can be approached for formal recruitment (if appropriate).

## Other ethical issues

Participants will be informed that they are free to withdraw from the study at any time, without needing to provide reason. At each data collection visit, the capacity of the participant will be checked (using an Abbreviated Mental Test) and asked if they are happy to proceed with data collection. Any concerns for a participant by the study team will be fed back to clinical staff. All blood samples will be collected by hospital staff and the research nurse team and will be stored, tested and disposed of in accordance with current United Kingdom guidelines and regulations. In the event of death within 3 months of being recruited, participants will be automatically withdrawn from the study and the primary predictive analysis. Baseline characteristics of withdrawn participants will be compared with those of retained participants to assess for any differences.

**Contributors** DF is Chief Investigator and guarantor. DF and ABR led protocol development, data analysis and dissemination. DWE is the Research Fellow with responsibility for study management. NM is a Doctoral Researcher focused to this study. NRH is the lead for patient and public involvement. JMP is the Principal Investigator at the Major Trauma Centre. JMP and CS are clinical representatives at the Major Trauma Centre. JL is the lead for biomarker evaluation. All authors will contribute to data interpretation, conclusions and dissemination. ABR drafted the initial manuscript with DF. Subsequent drafts were developed with DWE. All reviewers have read, contributed to and agreed upon the final manuscript.

**Funding** This work was supported by the National Institute for Health Research Surgical Reconstruction and Microbiology Research Centre (NIHR SRMRC) to DF and ABR.

**Competing interests** None declared.

**Patient consent** Not required.

**Ethics approval** Approved by the NHS Research Ethics Committee (17/WA/0421).

**Provenance and peer review** Not commissioned; externally peer reviewed.

**Data sharing statement** No additional data are available.

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
