## [Reviewer comments · BMJ Open]

ARTICLE DETAILS

TITLE (PROVISIONAL)	DEVELOPMENT OF A SCREENING TOOL TO PREDICT THE RISK OF CHRONIC PAIN AND DISABILITY FOLLOWING MUSCULOSKELETAL TRAUMA: PROTOCOL FOR A PROSPECTIVE OBSERVATIONAL STUDY IN THE UNITED KINGDOM
AUTHORS	Rushton, Alison; Evans, David; Middlebrook, Nicola; Heneghan, Nicola; Small, Charlotte; Lord, Janet; Patel, Jaimin; Falla, Deborah

VERSION 1 – REVIEW

REVIEWER	Steven George Duke University USA
REVIEW RETURNED	28-Jun-2017

GENERAL COMMENTS	Thank you for the chance to review this protocol paper. The authors have selected to focus on what factors are predictive of the acute to chronic pain transition. This is a topic of importance, and their proposed approach of including injury location, self-report, quantitative sensory, and biomarker in predictive models would progress the field. I realize this is a protocol paper, so will keep my reviewer comments based in areas that I think more rationale or description is needed to have the reader better understand why a given methodological choice was made. These comments are summarized below: 1. Methods/analysis, page 7 - it is not clear to this reviewer how the cohorts will be “temporally staggered”. Please provide detail on how this staggering occurs - does this mean that right after one cohort is completed, the next one will start? Clarity here would help the reader understand the overall recruitment approach better.2. Participants, page 8 - will there be any consideration at all of pre-trauma pain history in the eligibility criteria? I know from Table 1 that this will be considered from patient history and medical records, but it is not clear if recruitment is based on pre trauma pain history or if this information will be collected for covariates. In other words - will participants that have pre-existing chronic pain condition be included in these cohorts?3. Outcome, page 10 - It is not clear to this reviewer why the primary outcome (CPGS) is collected at 6 months, when there will be data collection at 12 months. The chronic pain status at 12 months would seem to match more definitions of “chronic” (this reviewer fully acknowledges there is no universal accepted definition but there is a trend for longer outcome tracking for pain outcomes is better) and be more in line with higher impact studies in this area. Please provide
--

	rationale for the primary endpoint at 6 months when 12-month collection is being completed. 4. Data handling, Table 1 and page 14 - there are some very important variables (e.g. previous history of musculoskeletal pain, comorbidity, and premorbid psychological and physical health) that are being collected by “patient history data from patient recollection and medical records”. There is no description of whether this process will be standardized or not, and what will happen if this information is missing from the record. These variables are likely to be important covariates and therefore this protocol paper would benefit from a better description of how this information will be collected, and what standard approaches will be used to ensure consistency in collection and minimizing loss of data. 5. Sample size, page 15 - the authors present an approach where they will adhere to 10 cases per candidate predictor (for a total of 250 subjects and 25 predictors) to power the study. However, there is no indication in the paper on how the 25 predictors will be determined. By my count there are at least 38 candidate variables from Table 1, and since many of those rows contain multiple data elements the actual number is much higher. The authors should provide some information on how these 25 predictors will be determined, otherwise there is risk this study will be underpowered. 6. Statistical analysis, page 15 - the authors have indicated the CPGS as primary outcome measure, which is ordinal scale. However, they propose use of logistic and linear regression. The authors should provide some justification for use of linear regression for the primary outcome, since other methods may be more appropriate for the scale of the measure. 7. Statistical analysis, page 15 - the authors indicate that initially analysis will “include all candidate prognostic factors and full results reported” and then reduced models will be considered. Rationale for this approach would help readers better understand this approach as the study does not seem adequately powered to address ALL the candidate predictors, and the way they are selected and reduced models are created should be provided in the protocol paper to avoid selective reporting of the final results. 8. Risk groups and development of screening tool, page 16 - the statement “those deemed clinically important to retain (regardless of statistical significance” to improve face validity for clinicians” needs to better described and rationale provided. Ideally these variables would already be identified and not contingent on statistical analysis. 9. Discussion, page 17 - the authors should provide more rationale on why they selected specific predictors beyond just indicating the four main categories. Some of this information may already be available from the supplementary material (I didn't review that part in detail) and could be incorporated into the actual paper. But readers unfamiliar with the pain research field will benefit from some brief rationale on why these specific measures were selected, and why there was only one biomarker selected.
--	---

REVIEWER	Mélanie Bérubé Ingram School of Nursing, McGill University, Montreal, Canada Centre Intégré Universitaire du Nord-de-l'Île de Montréal - Installation Hôpital du Sacré-Coeur de Montréal, Montreal, Canada
REVIEW RETURNED	05-Aug-2017
GENERAL COMMENTS	Comments:

This paper covers a very important topic. Chronic pain is a highly prevalent issue after a musculoskeletal trauma and developing a screening tool to identify patients at risk is an important step before developing relevant interventions.

However, they are important methodological issues that need to be clarified before proceeding with the publication of this paper:

- Musculoskeletal trauma should be better defined. This category of trauma is very large, from shoulder luxation to multiple fractures. Even though the study will be conducted in a major trauma center, this does not mean that only patients with severe musculoskeletal trauma will be recruited. The trauma population is very heterogeneous. In this regard, patients may have an abdominal trauma combined with ankle sprain. Will this patient be included? The patient with an ankle sprain is less likely to develop debilitating chronic pain than patients with multiple fractures. Including patients with ankle sprain might influence the validity of the tool. Including only patients with major musculoskeletal trauma (patients at risk of impaired outcomes usually requiring surgical and multidisciplinary team management) could be an alternative. I suggest involving an orthopaedic surgeon to help the research team determine who patients with such injury are.

- It is not clear if patients will be recruited at 14 days post-trauma or if patients will be recruited until 14 days post-trauma. If patients are recruited at 14 days post-trauma, several of them will have been already discharged. In the trauma center where I work, the median hospital length of stay of patients with major orthopaedic trauma is 9 days. Hence, recruiting patients at 14 days post-injury could impair the feasibility of study, unless patients' median hospital length of stay in the trauma center where the study will be conducted is longer than 14 days. Such information should be provided.

- Exclusion criteria should be better defined. The authors mentioned that they will exclude patients whom primary injury is not musculoskeletal providing the example of head injury. What about patients with abdominal trauma or a lung contusion and musculoskeletal injury? Will these patients be excluded? Isolated musculoskeletal injury is not that frequent in major trauma centers. If only these patients are targeted, it should be clearly stated and the number of concerned patients in the trauma center where the study will be conducted should be provided in order to better support the feasibility of the study.

Moreover, the authors mentioned that they will exclude patients with evident brain injury and patients with a GCS \leq 13. Patient with a GCS of 14 and 15 may have a positive scan of the head, hence evident brain injury. According to the exclusion criteria, these patients won't be included in the study. To avoid confusion, I suggest to specify that patients with moderate-severe TBI (GCS \leq 13) and spinal cord injury patients will be excluded, which are the two most common central nervous system injuries in the trauma population.

Since the study concerns the development of musculoskeletal chronic pain, patients with previous musculoskeletal chronic pain should be, by principle, an exclusion criterion. The authors should explain why it has not been identified as an exclusion criterion.

- In the recruitment section, the authors mentioned that they

estimated the number of eligible patients based on feasibility data. What are these feasibility data and where are they coming from? The injury profile of the 1000 eligible trauma patients should be described. Are they all patients with isolated musculoskeletal trauma?

- To reduce the burden associated with the study, the authors mentioned that they will wait a 1 hour to obtain baseline data following the informed consent process. Several data will be collected at baseline. Early after the injury, the trauma patients have a limited attention span. From my research experience with collecting data in the orthopaedic trauma population, I suggest to split baseline data collection in two, - i.e., doing the baseline questionnaires with patients in the same day after obtaining consent and performing quantitative sensory testing the following day. Also, to decrease burden study, the use of the shortest available questionnaires form should be considered (e.g., SF 12 items instead than the 36 items form).

- The outcome is defined as pain and disability at six months. To determine pain at six months, patient will be asked to describe their pain intensity during this period of time. However, the six months period includes the acute pain period. Chronic pain, by definition, occurs from 3 months post-injury. So, patients should be asked to describe their pain intensity in the last 3 months or over a shorter period of time. In this regard, patient's condition after a musculoskeletal injury evolves rapidly from 3 months to 6 months. It might be more representative of the real presence of chronic pain to ask patients to describe their pain intensity in the last month than in the last 3 months.

- A very comprehensive list of candidate predictors has been selected. Underlying reasons for selecting these predictors have been well describe for psychological features and pain characteristics. More information should be provided to support the selection of patient's characteristics, quality of life and physical functioning (add references on the influence of sleep and brain injury on pain) and injury characteristics (e.g. the link between mechanical ventilation and the development of chronic pain) predictors. Also, I suggest pain acceptance as a psychological predictor. This concept has been recognized as a potential chronic pain protective factor.

- In the risk groups and development of the prognostic screening tool, the authors mentioned that items deemed clinically important based on face validity for clinicians will be retained regardless of clinical significance. More information should be provided on how this face validity will be determined. Who will be the clinicians involved, how many of them and what will be the process to determine the best model afterwards. Likewise, it is described that included only one questionnaire item will be tested to facilitate the use of the model in clinical practice. Questionnaires are developed and validated to evaluate a concept. Using only one item means that it's not the actual concept that will be evaluated in clinical practice. Testing questionnaires dimensions (e.g., rumination in the Pain Catastrophizing Scale), would be a less reductive approach.

- In the Data source section, it is stated that data will be measured at 3 months and 12 months post-injury to explore the clinical course of recovery following injury in the shorter and long term. This is a

	secondary objective of the study and should appear in the Objectives section. Also, more detail should be provided on why data will be collected at this time points, -i.e., what are the links with the purpose of the actual study. Other comments:  - Introduction: I doubt that pain is an appropriate experience following a traumatic injury considering the suffering associate to it and the fact that it may lead to chronic pain. I would reconsider this statement since it is not fitting with content of the article. - When does rehabilitation starts according to the authors. In acute care settings or rehabilitation centers? What they mean by personalized rehabilitation should be defined. - Line 43: Research data should be used instead opinion to determine that chronic pain cannot be entirely attributable to psychological factors. - In the aims of study, the same classification of predictors as presented in the Candidate predictors should be used to improve consistency in the terminology. - The SPIRIT checklist is available as a supplementary file. However, it is not mentioned that this checklist was used in the text and that is a supplementary file. - The withdrawal parts after the exclusion criteria should be moved in the Ethics section. - I would change the term “consent sheet” for “information and consent form”. - In the Data handling section: More details should be provided on how the order of data collection will be randomized to prevent order effect. The same is true concerning who will check completeness and cross-checked data for accuracy. A description of the kind of collected interventions received should also been described. - In the Discussion section (page 18), the authors referred to possible effective interventions. Examples of such possible interventions should be provided to help figure out how having access to a predictive tool could decrease the risk of chronic pain.
--	--

VERSION 1 – AUTHOR RESPONSE

7th October 2017

Dear Editor

Ref: Manuscript ID bmjopen-2017-017876 entitled "DEVELOPMENT OF A SCREENING TOOL TO PREDICT THE RISK OF CHRONIC PAIN AND DISABILITY FOLLOWING MUSCULOSKELETAL TRAUMA: PROTOCOL FOR A PROSPECTIVE OBSERVATIONAL STUDY"

Thank you for your constructive feedback on our manuscript. Please find our specific response to your requests and details of revisions made below.

We hope that our responses have addressed the required revisions to a satisfactory level, and we thank the reviewers for their time in reviewing the draft manuscript.

Yours sincerely

Professor Deborah Falla
On behalf of all authors

Reviewer's Comments (authors' responses in red)

Reviewer: 1

Reviewer Name: Steven George

Institution and Country: Duke University, USA

Thank you for the chance to review this protocol paper. The authors have selected to focus on what factors are predictive of the acute to chronic pain transition. This is a topic of importance, and their proposed approach of including injury location, self-report, quantitative sensory, and biomarker in predictive models would progress the field.

I realize this is a protocol paper, so will keep my reviewer comments based in areas that I think more rationale or description is needed to have the reader better understand why a given methodological choice was made. These comments are summarized below:

1. Methods/analysis, page 7 - it is not clear to this reviewer how the cohorts will be "temporally staggered". Please provide detail on how this staggering occurs - does this mean that right after one cohort is completed, the next one will start? Clarity here would help the reader understand the overall recruitment approach better.

We expect there to be an interval of at least 6 months between recruitment into the two cohorts. We have amended the manuscript to explicitly reflect this.

2. Participants, page 8 - will there be any consideration at all of pre-trauma pain history in the eligibility criteria? I know from Table 1 that this will be considered from patient history and medical records, but it is not clear if recruitment is based on pre trauma pain history or if this information will be collected for covariates. In other words - will participants that have pre-existing chronic pain condition be included in these cohorts?

We will be recruiting some patients with premorbid musculoskeletal pain. We will record the location and severity (intensity and temporality) of this using standardised clinical report forms, for descriptive purposes and to use as a covariate during our analyses.

3. Outcome, page 10 - It is not clear to this reviewer why the primary outcome (CPGS) is collected at 6 months, when there will be data collection at 12 months. The chronic pain status at 12 months would seem to match more definitions of "chronic" (this reviewer fully acknowledges there is no universal accepted definition but there is a trend for longer outcome tracking for pain outcomes is better) and be more in line with higher impact studies in this area. Please provide rationale for the primary endpoint at 6 months when 12-month collection is being completed.

Several epidemiological studies have used a 6 month period as a criterion for chronic pain. Indeed, the reviewer will likely be aware that the CPGS was specifically developed to evaluate pain and disability over this specific period. In addition, Merskey and Bogduk (1994) use this period as their preferred temporal definition for chronic pain (see page xi in: <http://www.iasp-pain.org/files/Content/ContentFolders/Publications2/FreeBooks/Classification-of-Chronic-Pain.pdf>) The purpose of the data measured at 12 months is to further understand the clinical course of recovery longer term.

4. Data handling, Table 1 and page 14 - there are some very important variables (e.g. previous history of musculoskeletal pain, comorbidity, and premorbid psychological and physical health) that are being collected by "patient history data from patient recollection and medical records". There is no description of whether this process will be standardized or not, and what will happen if this information is missing from the record. These variables are likely to be important covariates and therefore this protocol paper would benefit from a better description of how this information will be collected, and what standard approaches will be used to ensure consistency in collection and minimizing loss of data.

It is now clarified that 'All data collection will be standardised through protocols and clinical report forms. Any missing data will be clearly reported.

Covariates are however not appropriate to a prediction model and so have not been included as part of the analysis. To ensure the reader's understanding that this paper focuses on predictive rather than explanatory modelling, the wording of predictive/ predictor has now been used throughout.

5. Sample size, page 15 - the authors present an approach where they will adhere to 10 cases per candidate predictor (for a total of 250 subjects and 25 predictors) to power the study. However, there is no indication in the paper on how the 25 predictors will be determined. By my count there are at least 38 candidate variables from Table 1, and since many of those rows contain multiple data elements the actual number is much higher. The authors should provide some information on how these 25 predictors will be determined, otherwise there is risk this study will be underpowered.

Under sample and in the data analysis section, we have now elaborated to include: 'The number of predictors will be reduced using exploratory factor analysis. This process will ensure that the sample size provides at least 10 cases per candidate predictor, to adequately power the final regression analysis'.

6. Statistical analysis, page 15 - the authors have indicated the CPGS as primary outcome measure, which is ordinal scale. However, they propose use of logistic and linear regression. The authors should provide some justification for use of linear regression for the primary outcome, since other methods may be more appropriate for the scale of the measure.

Thank you for picking up on this issue that remained following our previous consideration of a different measure of outcome. Logistic regression will be used when the outcome is dichotomised into 'good' or 'poor' outcome, where we will utilise the ordinal grading system of the CPGS.

7. Statistical analysis, page 15 - the authors indicate that initially analysis will "include all candidate prognostic factors and full results reported" and then reduced models will be considered. Rationale for this approach would help readers better understand this approach as the study does not seem adequately powered to address ALL the candidate predictors, and the way they are selected and reduced models are created should be provided in the protocol paper to avoid selective reporting of the final results.

Further detail has now been provided in the data analysis section of the manuscript to detail how candidate factors will be managed throughout the analysis. We hope that this reassures you of our robust a priori planned analyses to avoid selective reporting of results.

8. Risk groups and development of screening tool, page 16 - the statement "those deemed clinically important to retain (regardless of statistical significance" to improve face validity for clinicians" needs to be better described and rationale provided. Ideally these variables would already be identified and not contingent on statistical analysis.

A reference has been included to support this statement that enables use of expert opinion to avoid over-fitting of the model. This has also been clarified in the text.

9. Discussion, page 17 - the authors should provide more rationale on why they selected specific predictors beyond just indicating the four main categories. Some of this information may already be available from the supplementary material (I didn't review that part in detail) and could be incorporated into the actual paper. But readers unfamiliar with the pain research field will benefit from some brief rationale on why these specific measures were selected, and why there was only one biomarker selected.

The supplementary file contains a detailed rationale as to why specific measures were chosen. In addition, we have used the findings from pain studies in primary care to provide guidance for our choices. Since submitting the manuscript, we have liaised with colleagues and can now increase the number of biomarkers collected. These will still relate to the mechanistic categories used. The supplementary file has been extended accordingly but maintained as a supplementary file owing to its extensive nature.

Reviewer: 2

Reviewer Name: Mélanie Bérubé

Institution and Country: Ingram School of Nursing, McGill University, Montreal, Canada,

Comments:

This paper covers a very important topic. Chronic pain is a highly prevalent issue after a musculoskeletal trauma and developing a screening tool to identify patients at risk is an important step before developing relevant interventions.

Thank you, we appreciate this comment.

However, they are important methodological issues that need to be clarified before proceeding with the publication of this paper:

Thank you for these suggestions.

- Musculoskeletal trauma should be better defined. This category of trauma is very large, from shoulder luxation to multiple fractures. Even though the study will be conducted in a major trauma center, this does not mean that only patients with severe musculoskeletal trauma will be recruited. The trauma population is very heterogeneous. In this regard, patients may have an abdominal trauma combined with ankle sprain. Will this patient be included? The patient with an ankle sprain is less likely to develop debilitating chronic pain than patients with multiple fractures. Including patients with ankle sprain might influence the validity of the tool. Including only patients with major musculoskeletal trauma (patients at risk of impaired outcomes usually requiring surgical and multidisciplinary team

management) could be an alternative. I suggest involving an orthopaedic surgeon to help the research team determine who patients with such injury are.

We understand the point raised. We will be recording the tissue type and location that is injured and the timing of such injuries. However, from a mechanistic point of view, all of the above-mentioned injuries will produce nociception. In addition, there will likely be a strong post-trauma inflammatory response, and possible nerve injury. This mechanistic classification should enable us to combine injuries in different locations and tissues, and we have selected several of our candidate predictors around this (e.g. painDETECT questionnaire, QST, biomarkers).

- It is not clear if patients will be recruited at 14 days post-trauma or if patients will be recruited until 14 days post-trauma. If patients are recruited at 14 days post-trauma, several of them will have been already discharged. In the trauma center where I work, the median hospital length of stay of patients with major orthopaedic trauma is 9 days. Hence, recruiting patients at 14 days post-injury could impaired the feasibility of study, unless patients' median hospital length of stay in the trauma center where the study will be conducted is longer than 14 days. Such information should be provided.

We will be recruiting up to 14 days post-trauma (inclusive). We have amended the manuscript to make this explicit.

- Exclusion criteria should be better defined. The authors mentioned that they will exclude patients whom primary injury is not musculoskeletal providing the example of head injury. What about patients with abdominal trauma or a lung contusion and musculoskeletal injury? Will these patients been excluded? Isolated musculoskeletal injury is not that frequent in major trauma centers. If only these patients are targeted, it should be clearly stated and the number of concerned patients in the trauma center where the study will be conducted should be provided in order to better support the feasibility of the study.

A decision was made to exclude head injuries as this would be expected to confound pain assessment (both self-reported and QST measures) for at least the short-term. We are aware that concomitant organ injury is common in musculoskeletal trauma, but the long-term effect of this on pain is unknown, so we chose not to exclude these. We will be recording the occurrence of injuries occurring in other organs, which we can explore in secondary analyses. The exclusion criteria section has been reworded for greater clarity.

Moreover, the authors mentioned that they will exclude patients with evident brain injury and patients with a GCS \leq 13. Patient with a GCS of 14 and 15 may have a positive scan of the head, hence evident brain injury. According to the exclusion criteria, these patients won't be included in the study. To avoid confusion, I suggest to specify that patients with moderate-severe TBI (GCS \leq 13) and spinal cord injury patients will be excluded, which are the two most common central nervous system injuries in the trauma population.

We have amended the manuscript to reflect the above suggestion.

Since the study concerns the development of musculoskeletal chronic pain, patients with previous musculoskeletal chronic pain should be, by principle, an exclusion criterion. The authors should explain why it has not been identified as an exclusion criterion.

We will be asking patients about their premorbid pain history. Instead of excluding on this basis, we intend to use this factor as a candidate predictive factor within the analysis as we feel it may be an important factor in this population.

- In the recruitment section, the authors mentioned that they estimated the number of eligible patients based on feasibility data. What are these feasibility data and where are they coming from? The injury profile of the 1000 eligible trauma patients should be described. Are they all patients with isolated musculoskeletal trauma?

We used data for the Major Trauma Centre from the UK Trauma Audit and Research Network (TARN - www.tarn.ac.uk) database to provide these estimates. We have added this detail into the manuscript. We defined musculoskeletal trauma as that with no significant head injuries, which we excluded from our estimates.

- To reduce the burden associated with the study, the authors mentioned that they will wait a 1 hour to obtain baseline data following the informed consent process. Several data will be collected at baseline. Early after the injury, the trauma patients have a limited attention span. From my research experience with collecting data in the orthopaedic trauma population, I suggest to split baseline data collection in two, - i.e., doing the baseline questionnaires with patients in the same day after obtaining consent and performing quantitative sensory testing the following day.

Thank you - we agree that this burden should be split. We will now be collecting self-reported baseline data (via questionnaire) separately from (and at least 24 hours prior to) QST data.

- Also, to decrease burden study, the use of the shortest available questionnaires form should be considered (e.g., SF 12 items instead than the 36 items form).

The SF-36 was selected over the SF-12 because it includes a greater range of physical items (including transfers and mobility), which can be analysed as a separate subscale. We feel that this is important for this broad trauma population.

- The outcome is defined as pain and disability at six months. To determine pain at six months, patient will be asked to describe their pain intensity during this period of time. However, the six months period includes the acute pain period. Chronic pain, by definition, occurs from 3 months post-injury. So, patients should be asked to describe their pain intensity in the last 3 months or over a shorter period of time. In this regard, patient's condition after a musculoskeletal injury evolves rapidly from 3 months to 6 months. It might be more representative of the real presence of chronic pain to ask patients to describe their pain intensity in the last month than in the last 3 months.

There is debate over what is the most appropriate definition of chronic pain. We have used the Von Korff definition, which uses a frequency of painful and disabling days over a fixed period of time (e.g. 6 months), as this is favoured by epidemiologists. We used 6 months as against 1 month as this should capture a difference between those with a good and a poor outcome based on current understanding.

- A very comprehensive list of candidate predictors has been selected. Underlying reasons for selecting these predictors have been well described for psychological features and pain characteristics. More information should be provided to support the selection of patient's characteristics, quality of life and physical functioning (add references on the influence of sleep and brain injury on pain) and injury characteristics (e.g. the link between mechanical ventilation and the development of chronic pain) predictors. Also, I suggest pain acceptance as a psychological predictor. This concept has been recognized as a potential chronic pain protective factor.

Thank you - we have added more information for each of the candidate predictors raised.

We have given much thought to our list of candidate predictors, based on the existing pain literature in both trauma and primary care populations. We would have liked to include other factors e.g. pain

acceptance as candidate predictors, but we already have a large amount of data to collect and following discussion with our patient advisers have limited the data we are collecting to be mindful of participant burden.

- In the risk groups and development of the prognostic screening tool, the authors mentioned that items deemed clinically important based on face validity for clinicians will be retained regardless of clinical significance. More information should be provided on how this face validity will be determined. Who will be the clinicians involved, how many of them and what will be the process to determine the best model afterwards.

Likewise, it is described that included only one questionnaire item will be tested to facilitate the use of the model in clinical practice. Questionnaires are developed and validated to evaluate a concept. Using only one item means that it's not the actual concept that will be evaluated in clinical practice. Testing questionnaires dimensions (e.g., rumination in the Pain Catastrophizing Scale), would be a less reductive approach.

A reference has been included to support this statement that enables use of expert opinion to avoid over-fitting of the model. This has also been clarified in the text. It has also been clarified that this expert opinion will be afforded through the study steering group.

Regarding the issue of item versus questionnaire, we understand the issue raised. However, this methodological approach of item reduction to form a screening tool has been used successfully in low back pain research, and has been detailed by Hill et al (2008).

- In the Data source section, it is stated that data will be measured at 3 months and 12 months post-injury to explore the clinical course of recovery following injury in the shorter and long term. This is a secondary objective of the study and should appear in the Objectives section. Also, more detail should be provided on why data will be collected at this time points, -i.e., what are the links with the purpose of the actual study.

Thank you for this suggestion. We have now added this to the objectives section of the manuscript. We are interested in tracking the clinical course as previous longitudinal assessments of pain in primary care have shown that long-term pain 'trajectories' are typically established within the first months of a painful episode (Kongsted et al 2016).

Other comments:

- Introduction: I doubt that pain is an appropriate experience following a traumatic injury considering the suffering associate to it and the fact that it may lead to chronic pain. I would reconsider this statement since it is not fitting with content of the article.

We state that "Pain is an expected and appropriate experience following traumatic musculoskeletal injury." This is true given that traumatic injury is a noxious stimulus, and will induce nociception, from which we can expect a painful experience. This painful experience is appropriate since those who cannot experience pain from noxious stimuli (e.g. congenital analgesia) typically cannot attend to their injuries and typically suffer a reduced lifespan as a result (Nagasako et al 2003). The point we are trying to make is that short-term pain and disability, in line with tissue healing timetables, is appropriate and adaptive (useful), but long-term chronic pain and disability is not.

- When does rehabilitation starts according to the authors. In acute care settings or rehabilitation centers? What they mean by personalized rehabilitation should be defined.

When the phrase precision rehabilitation is first used it is now followed with the definition of 'identifying which patients to target with rehabilitation and when and how to target them'. This is inclusive of rehabilitation in all settings.

- Line 43: Research data should be used instead opinion to determine that chronic pain cannot be entirely attributable to psychological factors.

There is currently a lack of prognostic research to demonstrate much beyond psychological factors. However, this is precisely what our study is attempting to assess. Several experts have made calls for this, and we are responding to these.

- In the aims of study, the same classification of predictors as presented in the Candidate predictors should be used to improve consistency in the terminology.

This has been amended under the aims of the study.

- The SPIRIT checklist is available as a supplementary file. However, it is not mentioned that this checklist was used in the text and that is a supplementary file.

The SPIRIT checklist was required by the journal at time of submission. It is however not referred to in the text as the reviewers will acknowledge that it does not apply to this study, and hence it is not referred to in the text. We have referred to the TRIPOD statement (page 7) as several items of it could be used to inform this protocol, but have not included the checklist as all items could not be addressed prior to conducting the study.

- The withdrawal parts after the exclusion criteria should be moved in the Ethics section.

We have mentioned these here so that readers are made aware early on in the paper of these key issues. As recommended, we have also now mentioned this in the ethics section.

- I would change the term "consent sheet" for "information and consent form".

We have used the terminology of standardised forms in the UK; 'consent form' and 'participant information sheet'. We are expected to follow these conventions (for examples, see <http://www.hra-decisiontools.org.uk/consent/examples.html>).

- In the Data handling section: More details should be provided on how the order of data collection will be randomized to prevent order effect. The same is true concerning who will check completeness and cross-checked data for accuracy. A description of the kind of collected interventions received should also been described.

It is now detailed that randomisation will be achieved using online randomisation software.

The following has been added to confirm details of data checking:

'Data will be checked for completeness and accuracy following data entry by a second researcher. In addition, regular audits of data collection and storage will be performed by an independent study steering committee.

- In the Discussion section (page 18), the authors referred to possible effective interventions. Examples of such possible interventions should be provided to help figure out how having access to a predictive tool could decrease the risk of chronic pain.

There are many rehabilitation interventions. However, we feel it is too early to commit to any of these prior to data collection.

Overall further comments

Following collaboration with further colleagues to support, for example the extended set of biomarkers in this study, three further co-authors have been added to support delivery of this study.

The inclusion of further biomarkers as suggested has necessitated an amendment to the 'strengths and limitations of this study' section as the biomarkers are no longer a limitation.

The inclusion of further biomarkers has necessitated revision of the process of consent that has been revised accordingly. Specifically, the Mental Capacity Act 2005 is now referred to.

The review of the process of consent has also necessitated review of the eligibility criteria to particularly lead to our inclusion of the Abbreviated Mental Test.

References

Hill JC, Dunn KM, Lewis M, Mullis R, Main CJ, Foster NE, et al. A primary care back pain screening tool: identifying patient subgroups for initial treatment. *Arthritis Rheum* 2008;59(5):632–41.

Kongsted A, Kent P, Axen I, Downie AS, Dunn KM. What have we learned from ten years of trajectory research in low back pain? *BMC Musculoskelet Disord*. 2016 May 21;17:220.

Merskey H BN. Classification of chronic pain : descriptions of chronic pain syndromes and definitions of pain terms. USA: IASP Press 1994.

Nagasako EM, Oaklander AL, Dworkin RH. Congenital insensitivity to pain: an update. *Pain*. 2003 Feb;101(3):213-9.

VERSION 2 – REVIEW

REVIEWER	Steven George Duke University USA
REVIEW RETURNED	24-Oct-2017

GENERAL COMMENTS	I thank the authors for being responsive to original reviewer comments. The revised version of the paper is clearer on the methods that will be used to address this important research question. I have no further review suggestions and look forward to seeing these results published.
--

REVIEWER	Mélanie Bérubé McGill University, Montreal, Canada
REVIEW RETURNED	09-Nov-2017

GENERAL COMMENTS	1- In the Exclusion criteria section, page 10: The GCS is a well known assessment tool in trauma. I am not sure that it is required to provide a definition of this tool. However, it is not clear why the authors chose to exclude patients with a GCS ≤ 14. Patients with
---

	mild TBI have GCS between 13 and 15, not between 14 and 15. From a clinical point of view, I don't see any reason to exclude patients with a GCS ≤ 14 instead than ≤ 13. 2- In the Mental capacity section, page 24: There are repetitive information with those presented in the recruitment section concerning how data will be managed if patients regain or not capacity within 14 days following the trauma. 3- My major concern is related to the selection of the predictors. The rationale that led the selection of predictors is not always clear. For example, more explanations should be provided on the link between the time of injury/incident, employment status (might be better to use socio-economical status than employment status) and assisted mechanical ventilation with chronic pain. Pain catastrophizing and eligibility for compensation have also been identified as important predictors of chronic pain post orthopaedic trauma. In this regard, some key references to support the selection of predictors are not used and I would recommend the authors to consult them and add them to the references list to demonstrate that a rigorous review of literature was made to select predictors : Archer, K. R., Castillo, R. C., Wegener, S. T., Abraham, C. M., & Obremsky, W. T. (2012). Pain and satisfaction in hospitalized trauma patients: the importance of self-efficacy and psychological distress. Journal of Trauma and Acute Care Surgery, 72, 1068-1077.; Clay, F. J., Newstead, S. V., Watson, W. L., Ozanne-Smith, J., Guy, J., & McClure, R. J. (2010). Bio-psychosocial determinants of persistent pain 6 months after non-life-threatening acute orthopaedic trauma. Journal of Pain, 11, 420-430.; Clay, F. J., Watson, W. L., Newstead, S. V., & McClure, R. J. (2012). A systematic review of early prognostic factors for persisting pain following acute orthopedic trauma. Pain Research & Management, 17(1), 35-44.; Katz, J., & Seltzer, Z. (2009). Transition from acute to chronic postsurgical pain: risk factors and protective factors. Expert Review of Neurotherapeutics, 9, 723-744.; Rosenbloom, B. N., Khan, S., McCartney, C., & Katz, J. (2013). Systematic review of persistent pain and psychological outcomes following traumatic musculoskeletal injury. Journal of Pain Research, 6, 39-51.; Wegener, S. T., Castillo, R. C., Haythornthwaite, J., Mackenzie, E. J., Bosse, M. J., & Leap Study Group. (2011). Psychological distress mediates the effect of pain on function. Pain, 152, 1349-1357.; Rosenbloom, B.N., et al., Predicting pain outcomes after traumatic musculoskeletal injury. Pain, 2016. 157(8): p. 1733-43.; Vranceanu, A. M., Bachoura, A., Weening, A., Vrahas, M., Smith, R. M., & Ring, D. (2014). Psychological factors predict disability and pain intensity after skeletal trauma. J Bone Joint Surg Am, 96(3), e20.
--	--

VERSION 2 – AUTHOR RESPONSE

Dear Editor

Thank you for your constructive feedback on our manuscript. Please find our specific response to your requests and details of revisions made below.

We hope that our responses have addressed the required revisions to a satisfactory level, and we thank the reviewers for their time in reviewing the draft manuscript.

Yours sincerely,

Professor Deborah Falla
On behalf of all authors

Reviewer's Comments – second submission

Reviewer: 1

Reviewer Name: Steven George
Institution and Country: Duke University, USA
Please state any competing interests: None declared

Please leave your comments for the authors below

I thank the authors for being responsive to original reviewer comments. The revised version of the paper is clearer on the methods that will be used to address this important research question. I have no further review suggestions and look forward to seeing these results published.

The authors would like to thank Reviewer 1 for his time spent reviewing our work and useful commentary.

Reviewer: 2

Reviewer Name: Mélanie Bérubé
Institution and Country: Ingram School of Nursing, McGill University, Montreal, Canada, Centre Intégré Universitaire du Nord-de-l'Île de Montréal - Installation Hôpital du Sacré-Coeur de Montréal, Montreal, Canada
Please state any competing interests: None to declare

Please leave your comments for the authors below

Comments:

1- In the Exclusion criteria section, page 10: The GCS is a well known assessment tool in trauma. I am not sure that it is required to provide a definition of this tool. However, it is not clear why the authors chose to exclude patients with a GCS ≤ 14 . Patients with mild TBI have GCS between 13 and 15, not between 14 and 15. From a clinical point of view, I don't see any reason to exclude patients with a GCS ≤ 14 instead than ≤ 13 .

As this is a protocol paper in a generic biomedical journal, we thought it appropriate to provide a description of the GCS (as below) as it clarifies what it measures, that it is reliable (we have also described the psychometric properties of all other measures we are using, even those commonly utilised), and chronologically when it will be measured in our participants, so that we can justify why we have chosen it amongst our exclusion criteria: "a 15-item measure of consciousness impairment with adequate reliability that is routinely taken in trauma patients at hospital admission."

The exclusion criterion using a GCS ≤ 14 , combined with any intra-cranial lesion was set by our co-investigator, an experienced Consultant in Anaesthesia and Intensive Care Medicine at a Major

Trauma Centre, to reflect clinical practice decision-making. The aim was to define an exclusion criterion that is very sensitive to consciousness impairment (which would otherwise threaten the validity of our baseline measures), requiring only 1 point loss on the GCS, but specific in that it relates directly to compromising pathology of the central nervous system.

2- In the Mental capacity section, page 24: There are repetitive information with those presented in the recruitment section concerning how data will be managed if patients regain or not capacity within 14 days following the trauma.

We accept that there is a small amount of repetition; however, as a protocol paper we felt that this was necessary to ensure the reader was aware of our recruitment and data collection procedures for those that lacked mental capacity upon admission and prior to 14 days post-injury.

3- My major concern is related to the selection of the predictors. The rationale that led the selection of predictors is not always clear. For example, more explanations should be provided on the link between the time of injury/incident, employment status (might be better to use socio-economical status than employment status) and assisted mechanical ventilation with chronic pain. Pain catastrophizing and eligibility for compensation have also been identified as important predictors of chronic pain post orthopaedic trauma. In this regard, some key references to support the selection of predictors are not used and I would recommend the authors to consult them and add them to the references list to demonstrate that a rigorous review of literature was made to select predictors :

Archer, K. R., Castillo, R. C., Wegener, S. T., Abraham, C. M., & Obremskey, W. T. (2012). Pain and satisfaction in hospitalized trauma patients: the importance of self-efficacy and psychological distress. *Journal of Trauma and Acute Care Surgery*, 72, 1068-1077.

Clay, F. J., Newstead, S. V., Watson, W. L., Ozanne-Smith, J., Guy, J., & McClure, R. J. (2010). Bio-psychosocial determinants of persistent pain 6 months after non-life-threatening acute orthopaedic trauma. *Journal of Pain*, 11, 420-430.

Clay, F. J., Watson, W. L., Newstead, S. V., & McClure, R. J. (2012). A systematic review of early prognostic factors for persisting pain following acute orthopedic trauma. *Pain Research & Management*, 17(1), 35-44.

Katz, J., & Seltzer, Z. (2009). Transition from acute to chronic postsurgical pain: risk factors and protective factors. *Expert Review of Neurotherapeutics*, 9, 723-744.

Rosenbloom, B. N., Khan, S., McCartney, C., & Katz, J. (2013). Systematic review of persistent pain and psychological outcomes following traumatic musculoskeletal injury. *Journal of Pain Research*, 6, 39-51.

Wegener, S. T., Castillo, R. C., Haythornthwaite, J., Mackenzie, E. J., Bosse, M. J., & Leap Study Group. (2011). Psychological distress mediates the effect of pain on function. *Pain*, 152, 1349-1357.

Rosenbloom, B.N., et al., Predicting pain outcomes after traumatic musculoskeletal injury. *Pain*, 2016. 157(8): p. 1733-43.

Vranceanu, A. M., Bachoura, A., Weening, A., Vrahas, M., Smith, R. M., & Ring, D. (2014).

Psychological factors predict disability and pain intensity after skeletal trauma. *J Bone Joint Surg Am*, 96(3), e20.

We are aware of the current literature and acknowledge that we could provide more detail in the paper for justifying the selection of specific candidate predictors. As such, we have now added more detail to the Supplemental File, and included most of the suggested studies amongst this.

We would like to thank Reviewer 2 for her time and useful comments.

VERSION 3 – REVIEW

REVIEWER	Melanie Berube Ingram School of Nursing, McGill University, Montreal, Canada, Centre IntégréUniversitaire du Nord-de-l'Île de Montréal Installation Hôpital du SacréCoeur de Montréal, Montreal, Canada
REVIEW RETURNED	17-Jan-2018
GENERAL COMMENTS	I thank the authors to have consider reviewer comments. I have no further review suggestions and look forward to seeing these results published.

VERSION 3 – AUTHOR RESPONSE

February 1, 2018

Dear Dr Groves,

Enclosed is our revised manuscript entitled “DEVELOPMENT OF A SCREENING TOOL TO PREDICT THE RISK OF CHRONIC PAIN AND DISABILITY FOLLOWING MUSCULOSKELETAL TRAUMA: PROTOCOL FOR A PROSPECTIVE OBSERVATIONAL STUDY IN THE UNITED KINGDOM” to be considered for publication in BMJ Open.

We confirm that ethical approval has now been granted by a NHS Research Ethics Committee (17/WA/0421) and this is now stated within the revised manuscript.

We have also added the location of the study to the title as per journal requirements.

Thank you again for considering this submission.

Yours sincerely,

Deborah Falla

::

Professor Deborah Falla
Chair in Rehabilitation Science and Physiotherapy

School of Sport, Exercise and Rehabilitation Sciences
College of Life and Environmental Sciences
University of Birmingham
Edgbaston B15 2TT
UK
T: +44 (0)121 41 47253
E: d.falla@bham.ac.u